# A rapid theta network mechanism for flexible information encoding

Elizabeth L. Johnson [1] ✉, Jack J. Lin[2], David King-Stephens[3,4], Peter B. Weber [3], Kenneth D. Laxer[3], Ignacio Saez [5,6], Fady Girgis[5,7], Mark D'Esposito [8], Robert T. Knight [8] & David Badre [9] ✉

Flexible behavior requires gating mechanisms that encode only task-relevant information in working memory. Extant literature supports a theoretical division of labor whereby lateral frontoparietal interactions underlie information maintenance and the striatum enacts the gate. Here, we reveal neocortical gating mechanisms in intracranial EEG patients by identifying rapid, within-trial changes in regional and inter-regional activities that predict subsequent behavioral outputs. Results first demonstrate information accumulation mechanisms that extend prior fMRI (i.e., regional high-frequency activity) and EEG evidence (inter-regional theta synchrony) of distributed neocortical networks in working memory. Second, results demonstrate that rapid changes in theta synchrony, reflected in changing patterns of default mode network connectivity, support filtering. Graph theoretical analyses further linked filtering in task-relevant information and filtering out irrelevant information to dorsal and ventral attention networks, respectively. Results establish a rapid neocortical theta network mechanism for flexible information encoding, a role previously attributed to the striatum.

Flexible behavior requires gating mechanisms that select goal- or task-relevant information to encode and maintain in an accessible, working state, while keeping irrelevant information out[1–3]. This accessible, working state is termed working memory (WM), and effective use of WM enables us to continuously update our goals based on incoming information. The prefrontal cortex (PFC) is critical to WM and positioned at the apex of distinct brain networks posited to underlie the gating and accumulation of information maintained in WM[4–11]. Converging neuroimaging, neuropharmacological, and neuropsychological evidence supports computational models that attribute gating to interactions between PFC and the striatum. In these models, the striatum is viewed as a gate that opens and closes, flexibly controlling

access to and from WM through reciprocal disinhibitory loops with PFC[1,2,12–18]. Maintenance in WM, however, is supported by cortical networks, with emphasis on interactions between PFC and parietal cortex[6,10,19,20]. By separating maintenance and gating between the neocortex and striatum, these models balance the fundamental computational tradeoff between stability and flexibility. It is unknown whether broader neocortical network dynamics, which feature sub-second activities distributed beyond the frontoparietal network, also contribute to flexible gating processes in WM.

We address this unknown by examining intracranial EEG (iEEG) data recorded simultaneously from PFC and distributed brain regions in neurosurgical patients as they performed an established task of WM

[1]Departments of Medical Social Sciences and Pediatrics, Northwestern University, Chicago, IL, USA. [2]Department of Neurology and Center for Mind and Brain, University of California, Davis, CA, USA. [3]Department of Neurology and Neurosurgery, California Pacific Medical Center, San Francisco, CA, USA. [4]Department of Neurology, Yale School of Medicine, New Haven, CT, USA. [5]Department of Neurological Surgery, University of California, Davis, CA, USA. [6]Departments of Neuroscience, Neurosurgery, and Neurology, Ichan School of Medicine at Mt. Sinai, New York, NY, USA. [7]Department of Clinical Neurosciences, University of Calgary, Calgary, AB, Canada. [8]Helen Wills Neuroscience Institute and Department of Psychology, University of California, Berkeley, CA, USA. [9]Department of Cognitive, Linguistic, and Psychological Sciences, and Carney Institute for Brain Science, Brown University, Providence, RI, USA. ✉e-mail: eljohnson@northwestern.edu; david_badre@brown.edu

gating previously used with fMRI[1,13,14]. This single-trial WM task is comprised of three sequentially presented and reorderable stimuli: two 'item' stimuli and a 'context' stimulus that specifies the relevant item (Fig. 1a). When the context appears first (CF), it can be used to drive all-or-none input gating of only the target item into WM. When the context appears last (CL), it can only be used for selectively output gating the target item from WM stores. Extant data from this task link correlated fMRI hemodynamic responses between PFC and the striatum to efficient output gating, as indexed by subsequent behavioral response times (RT)[1,13,14]. However, hemodynamic responses arise from multiple, mixed neurophysiological mechanisms[21,22] and, due to the temporal limitations of fMRI, sub-second changes in brain dynamics during the gating and accumulation of incoming information could not be investigated. iEEG provides spatial and temporal precision and high signal-to-noise ratio enabling single-trial reliability, making it uniquely suited to elucidate changing brain dynamics as stimuli are presented in sequence[23,24]. Because iEEG rarely, if ever, samples the striatum, our study addresses neocortical mechanisms of WM gating and makes no claim about striatal involvement. It is likely that neocortical involvement in WM gating is accompanied, or even influenced, by activity in the striatum.

Evidence that cortical neurophysiology supports the gating and accumulation of incoming information comes from a recent study in non-human primates reporting that the same PFC neurons are involved in 'attention and WM'[25]. In that study, attention was controlled using pre-cues in a task that is remarkably similar in design to our CF trials, and WM was controlled using retro-cues in a task that is also similar in design to our CL trials. It is thus reasonable to suppose that the same neuronal populations support both the gating and accumulation of incoming information in the human PFC. It remains an open question whether mechanisms that enact interactions across PFC and broader neocortical networks also support these processes, as would be evident in changes in inter-regional connectivity.

We analyzed local field potentials and inter-regional connectivity to address these questions[26]. Specifically, we tested a model in which neural oscillations, which provide optimal windows for excitability and inter-regional coordination[27], dually accumulate relevant information in and block irrelevant information from WM[3,11,28]. Based on converging theories of human cognitive control[29,30] and evidence from iEEG studies of attention and WM[23,24,31–37], we hypothesized that theta oscillations would serve this dual role. To test this hypothesis, we identified rapid mechanisms of information accumulation (i.e., encoding items under load) and filtering (encoding targets over distractors) based on trial-by-trial behavioral outcomes.

We first demonstrate mechanisms of information accumulation on CL trials that extend evidence from fMRI with regional high-frequency electrophysiological activity[38] and from scalp EEG with intracranial inter-regional theta synchrony[8,39,40] supporting distributed neocortical networks involved in WM gating. We then reveal inter-regional theta synchrony supporting filtering on CF trials, reflected in rapidly changing patterns of default mode network connectivity. Graph theoretical analyses further linked encoding or filtering in task-relevant information and blocking or filtering out irrelevant information to dorsal and ventral attention networks, respectively. Our results establish a rapid neocortical theta network mechanism for input gating that supports fast and flexible human behavior.

## Results

### Input gating manipulation and strategy use

Eleven neurosurgical patients participated (Table S1). Subjects were selected on the basis of above-chance behavioral performance across CF and CL trials and, accordingly, our sample performed well overall (M ± SD proportion errors, 0.17 ± 0.12; chance 0.5, $F(1,20) = 91.65$,

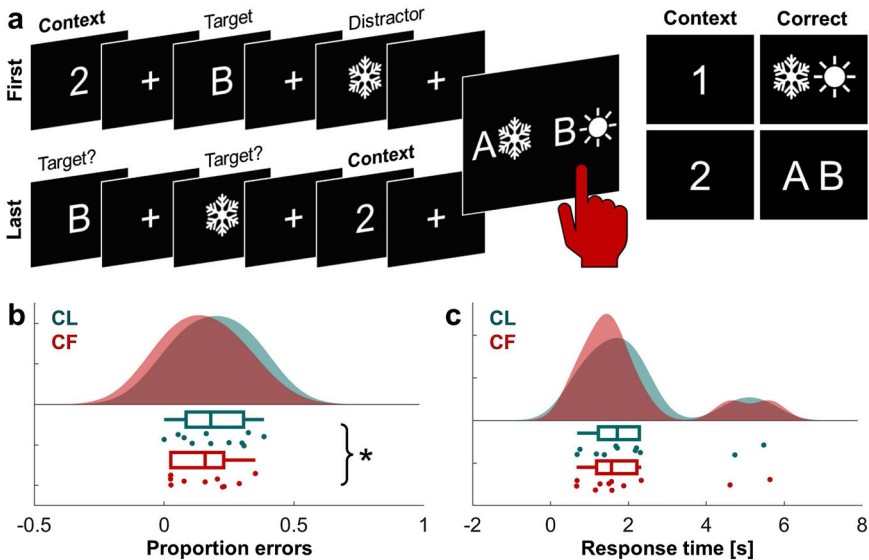

**Fig. 1 | Input gating manipulation and strategy use. a** Task schematic. Subjects completed a single-trial WM task comprised of three sequentially presented and reorderable stimuli: two 'item' stimuli (a letter and a symbol) and a 'context' stimulus (a number) that specified which item was relevant. When the context appeared first (CF; top), it could be used to drive input gating of only the relevant item into WM. When the context appeared last (CL; bottom), it could only be used for selectively output gating the relevant item. Each stimulus was on screen for 500 ms, followed by a randomly jittered inter-stimulus fixation between 250 and 800 ms. The trial concluded with response mappings, to which subjects had to indicate (by left or right button press) where the relevant item appeared. Inset: Correct responses by context. Numbers acted as higher-order context, specifying which of the lower-level items (two possible letters and two possible symbols) was relevant on each trial. CF, context first; CL, context last. **b** Superior performance accuracy (i.e., fewer errors) on CF compared to CL trials demonstrates that subjects tended to use an input gating strategy when possible ($F(1,20) = 6.83$, $p = 0.017$). Data are represented as individual datapoints, and condition probability densities and medians calculated across subjects ($n = 11$ biologically independent samples). Boxplots present the medians and interquartile ranges, and whiskers the 1.5*IQR from the quartile. Source data are provided as a Source Data file. *$p < 0.05$. **c** RT did not differ significantly between CF and CL trials ($n = 11$ biologically independent samples), same conventions as **b**.

$p = 7 \times 10^{-9}$). We next investigated condition differences. Subjects made significantly fewer errors on CF ($0.15 \pm 0.12$) compared to CL trials ($0.20 \pm 0.13$; $F(1,20) = 6.83$, $p = 0.017$, Cohen's $d = -0.75$; Fig. 1b). This difference is important, as subjects could have waited until the response screen to decide which item to select. Superior performance accuracy on CF trials demonstrates that subjects tended to adopt the proactive input gating strategy available to them on these trials and CL trials imposed a cost associated with load and selection from within WM[13].

Analysis of RT data indicated a non-significant cost associated with load and selection from within WM on CL trials (M ± SD CF, $2.08 \pm 1.59$ s; CL, $2.21 \pm 1.54$ s; $F(1,20) = 2.85$, $p = 0.107$, Cohen's $d = -0.49$; Fig. 1c). This non-significant result is consistent with prior work and may reflect an individual's tendency to select the target item during the brief epoch between presentation of the context and subsequent response screen (Fig. 1a)[13]. In the analysis of iEEG data, we minimized inter-individual differences while maintaining the logic of prior work using this task. To identify neurophysiological mechanisms supporting the flexible encoding of relevant information into WM, we capitalized on the single-trial reliability of iEEG[23] by quantifying within-trial shifts in iEEG measures based on subsequent behavioral accuracy and RT in subject-level analyses. Our approach builds on prior work with this task linking output gating to faster RTs on a per-subject basis in group-level analysis of fMRI data.

### High-frequency broadband mechanism of information encoding

High-frequency broadband (HFB) activity, a measure of power typically bounded between 70 and 150 Hz, tracks population-level neuronal activity[41–45] and hemodynamic responses[46,47]. HFB activity provides a powerful tool to assess the occurrence and magnitude of regional activity during task performance, and bridge fMRI to human neurophysiology[23]. We utilized established procedures to identify task-

responsive electrodes based on HFB activity across correct CF and CL trials[35,48–52], as detailed in the "Methods" section. Increased task-related HFB activity above baseline was detected in 32.3% (145 of 449) of bipolar montage-referenced electrodes (Fig. S1a and Table S1). This percentage is consistent with literature on task responsiveness across a range of cognitive tasks and brain regions[50]. Task-responsive electrodes, all of them artifact-free and nonpathological, were categorized by neocortical network and analyzed further. Electrodes were categorized into seven networks based on the Yeo Atlas: visual (VSN), somatomotor (SMN), dorsal attention (DAN), ventral attention (VAN), limbic (LBN), frontoparietal (FPN), and default mode (DMN)[53].

To identify neurophysiological mechanisms supporting the flexible encoding of relevant information into WM, we first determined mechanisms of information accumulation on CL trials. Specifically, we tested whether within-trial shifts in regional HFB activity predicted subsequent behavioral accuracy and RT. On each trial, we subtracted data from the epoch encompassing the encoding and maintenance of the first item from that of the second item, prior to the context. Specifically, we averaged HFB data over the 750-ms epoch extending from the onset of each encoding stimulus (500 ms on screen, 250 ms fixation; Fig. 1a), yielding two datapoints per trial from which we computed difference scores. We then separately averaged the difference scores across correct and error trials per subject, and correlated difference scores with behavioral RT across correct CL trials per subject. Correlation coefficients were then z-score normalized against null distributions. These procedures yielded one mean correct difference score, one mean error difference score, and one normalized correlation coefficient per electrode. Importantly, correlation analysis captured gating mechanisms without assuming that subjects employed gating strategies on all trials. A positive difference and/or negative correlation would indicate that a greater item 2 effect preceded a (faster) correct response, consistent with accumulating information in

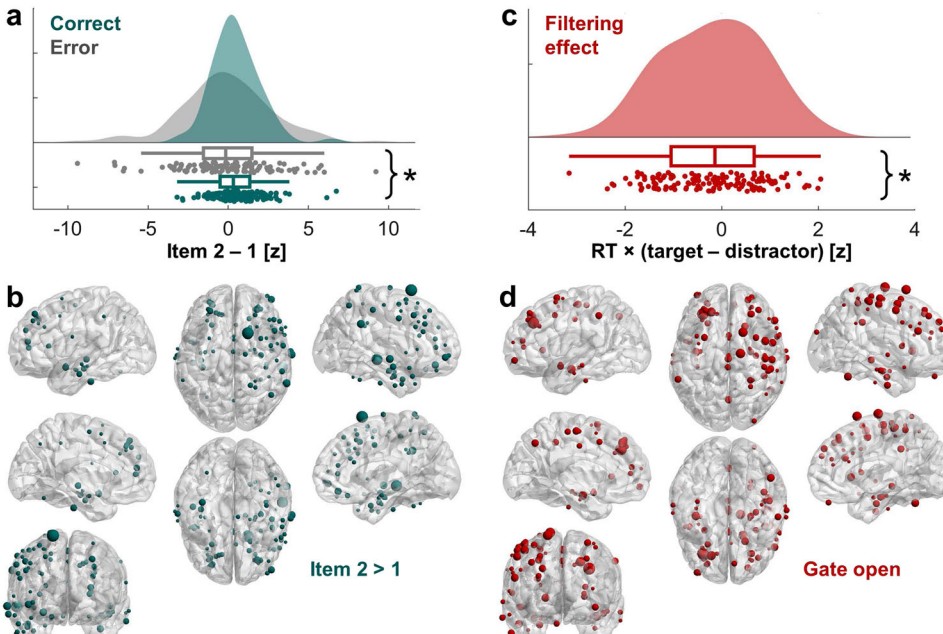

**Fig. 2 | Regional HFB information accumulation and filtering effects supporting successful WM. a** Successful CL performance was linked to increased HFB activity to the second over first item ($F(1,255) = 13.082$, $p = 0.0004$). Data are represented as individual electrode datapoints, and condition probability densities and medians calculated across electrodes from all subjects ($n = 145$ electrodes from 11 biologically independent samples). Boxplots present the medians and interquartile ranges, and whiskers the $1.5 \times$ IQR from the quartile. Source data are provided as a Source Data file. \*$p < 0.05$. **b** Electrodes contributing to the significant correct > error effect in (**a**), overlaid on the MNI-152 template brain. Electrode size

indicates the relative size of the item 2 > 1 difference on correct trials ($0 < z < 6.75$), consistent with information accumulation in WM. This figure was created using BrainNet Viewer[72]. **c** On correct CF trials, faster behavioral RT was linked to increased HFB activity to targets over distractors ($F(1,276) = 6.86$, $p = 0.009$; $n = 145$ electrodes from 11 biologically independent samples), same conventions as **a**. **d** Electrodes contributing to the significant RT correlation effect in **c** overlaid on the MNI-152 template brain. Electrode size indicates the relative size of the normalized RT × (target > distractor) correlation ($-3.16 < z < 0$), consistent with an open gate. This figure was created using BrainNet Viewer[72].

WM, and a negative difference and/or positive correlation that a greater item 1 effect preceded a (faster) correct response, consistent with a transition from encoding item 1 to item 2. Here, the input gate must be selective because each item is assigned to a slot in WM in a way that is addressable based on its role or category to be subsequently selected based on context[12,18,25]. This 'role addressability' is an essential characteristic of selective gating in computational models.

We implemented a linear mixed-effects model analysis to examine HFB difference scores as a function of behavioral accuracy (i.e., correct vs. error) and brain network, treating subjects and nested electrodes as random intercepts[54]. This group-level analysis revealed an accuracy main effect ($F(1,255) = 13.082$, $p = 0.0004$), with successful performance linked to increased HFB activity during the encoding of the second over first item (Fig. 2a). The network main effect and interaction were not significant ($p > 0.06$). HFB information accumulation effects were distributed across neocortical networks (Fig. 2b). The equivalent model examining HFB activity by behavioral RT (i.e., normalized correlations vs. zero[35]) and brain network did not reveal any significant effects ($p > 0.34$).

Next, we applied the same technique on data from CF trials to examine whether HFB activity also supports the encoding of task-relevant over irrelevant information. On these trials, the distractor stimulus was subtracted from the target stimulus, regardless of presentation order (Fig. 1a). We then separately averaged the difference scores across correct and error trials per subject, and correlated difference scores with behavioral RT across correct CF trials per subject. A positive difference and/or negative correlation would indicate that a greater target effect preceded a (faster) correct response, consistent with an 'open gate', and a negative difference and/or positive correlation that a greater distractor effect preceded a (faster) correct response, consistent with a 'closed gate'. We refer to this as filtering because it tests how well the input gate can distinguish targets from distractors in updating items to maintain in WM. In this case, the gate could be selective or global because updating of stimulus to memory is all-or-none.

The linear mixed-effects model examining HFB difference scores by accuracy and network did not reveal any significant effects ($p > 0.65$). However, the equivalent model examining HFB activity by RT and brain network revealed an RT main effect ($F(1,276) = 6.86$, $p = 0.009$; Fig. 2c). Faster RT was linked to increased HFB activity during the encoding of targets over distractors, consistent with an open gate. The network main effect and interaction were not significant ($p > 0.41$). Like information accumulation effects, HFB target encoding effects were distributed across neocortical networks (Fig. 2d).

Collectively, these results are consistent with fMRI evidence on the involvement of distributed brain regions in WM[38]. Results extend prior work by associating rapid changes in regional HFB activity directly to subsequent behavioral outputs and provide initial support for our hypothesis by demonstrating that regional HFB gating effects are distributed across neocortical networks. Our results suggest that HFB activity accumulates potentially relevant information in WM to drive successful performance, and filters in targets over distractors to drive efficient selection of relevant information from WM.

### Oscillatory characteristics of information encoding
To determine the oscillatory characteristics of task-responsive electrodes, we utilized an irregular sampling procedure that separates oscillatory components from the aperiodic $1/f$ slope[55]. Oscillatory peaks were categorized using canonical theta (4–8 Hz), alpha (8–13 Hz), and beta (13–30 Hz) bands (Fig. S1b-e). Peaks were detected in the theta band in 75.9% (110 of 145) of responsive electrodes. In contrast, alpha and beta peaks were detected in 34.5% (50 of 145; test of proportion vs. theta, $\chi^2 = 50.19$, $p = 10^{-12}$) and 23.5% (34 of 145; $\chi^2 = 79.67$, $p = 0$) of responsive electrodes, respectively. The prevalence of theta oscillations during information encoding across CF and CL

trials motivated analysis of encoding functions in theta networks. We additionally examined the smaller subsets of alpha and beta electrodes for completeness.

### Theta network mechanism of information accumulation
Unlike HFB activity, which tracks regional activity, neural oscillations underlie functional connectivity by coordinating periodic changes in the excitability of different neuronal populations[27]. According to the 'communication through coherence' hypothesis, the oscillatory phase relationship between regions dictates the strength of inter-regional communication[56]. We assessed phase relationships between pairs of electrodes exhibiting theta oscillations by means of phase-locking values (PLV)[57], and submitted the outputs to the same single-subject analyses and group-level models as HFB activity to examine inter-regional theta synchrony. For PLV, these procedures yielded one mean correct difference score, one mean error difference score, and one normalized correlation coefficient per electrode pair ($n = 1374$; Fig. S2).

We first identified information accumulation effects on CL trials. The linear mixed-effects model examining theta synchrony difference scores (i.e., item 2–1) as a function of behavioral accuracy and brain network revealed a brain network main effect ($F(36,920) = 2.16$, $p = 0.0001$) and an accuracy by network interaction ($F(1,920) = 1.54$, $p = 0.023$). The accuracy main effect was not significant ($p = 0.13$). Planned post hoc models tested for an accuracy effect across each pair of networks and within each network. Successful performance was linked to increased theta synchrony during the encoding of the second over first item in VAN-FPN ($t(88) = 2.27$, $p = 0.026$) and intra-LBN ($t(36) = 2.62$, $p = 0.013$) interactions (Fig. 3a–c). All post hoc results are provided in Fig. 3d. Successful performance was further linked to minimal change in theta synchrony between items in DMN-VAN interactions ($t(50) = -2.54$, $p = 0.014$; Fig. 3e, f). The equivalent model examining theta synchrony by RT and brain network did not reveal any significant effects ($p > 0.10$).

These results are consistent with EEG evidence of inter-regional functional connectivity in the theta band during WM[8,39,40]. Results extend prior work by associating rapid changes in inter-regional theta synchrony directly to subsequent behavioral outputs and provide further support for our hypothesis by demonstrating that theta oscillations accumulate task-relevant information in WM.

### Theta network mechanism of filtering
Next, we identified filtering effects on CF trials to examine whether theta oscillations also block irrelevant information from WM. The linear mixed-effects model examining theta synchrony difference scores (i.e., target – distractor) as a function of behavioral accuracy and brain network revealed a brain network main effect ($F(41,1290) = 2.23$, $p = 2 \times 10^{-5}$) and an accuracy by network interaction ($F(41,1290) = 1.52$, $p = 0.020$). The accuracy main effect was not significant ($p = 0.73$). Post hoc testing revealed that successful performance was linked to minimal change in theta synchrony between items in DMN-DAN interactions ($t(4) = -12.70$, $p = 0.0002$; Fig. 4a, b). All post hoc results are provided in Fig. 4c. No effects between or within other networks reached significance ($p > 0.07$).

The equivalent model examining theta synchrony by RT and brain network revealed an RT by network interaction ($F(41,1290) = 1.55$, $p = 0.016$). The RT and network main effects were not significant ($p > 0.23$). Post hoc models tested for an RT effect across each pair of networks and within each network. Faster RT was linked to increased theta synchrony during the encoding of targets over distractors in DMN-DAN ($t(4) = -8.55$, $p = 0.001$), FPN-DAN ($t(24) = -2.12$, $p = 0.045$), and intra-DAN ($t(4) = -5.14$, $p = 0.007$) interactions (Fig. 5a–d), consistent with an open gate. All post hoc results are provided in Fig. 5e. Faster RT was further linked to increased theta synchrony during the blocking of distractors in DMN-LBN interactions ($t(40) = 2.28$, $p = 0.028$; Fig. 5f, g), consistent with a closed gate.

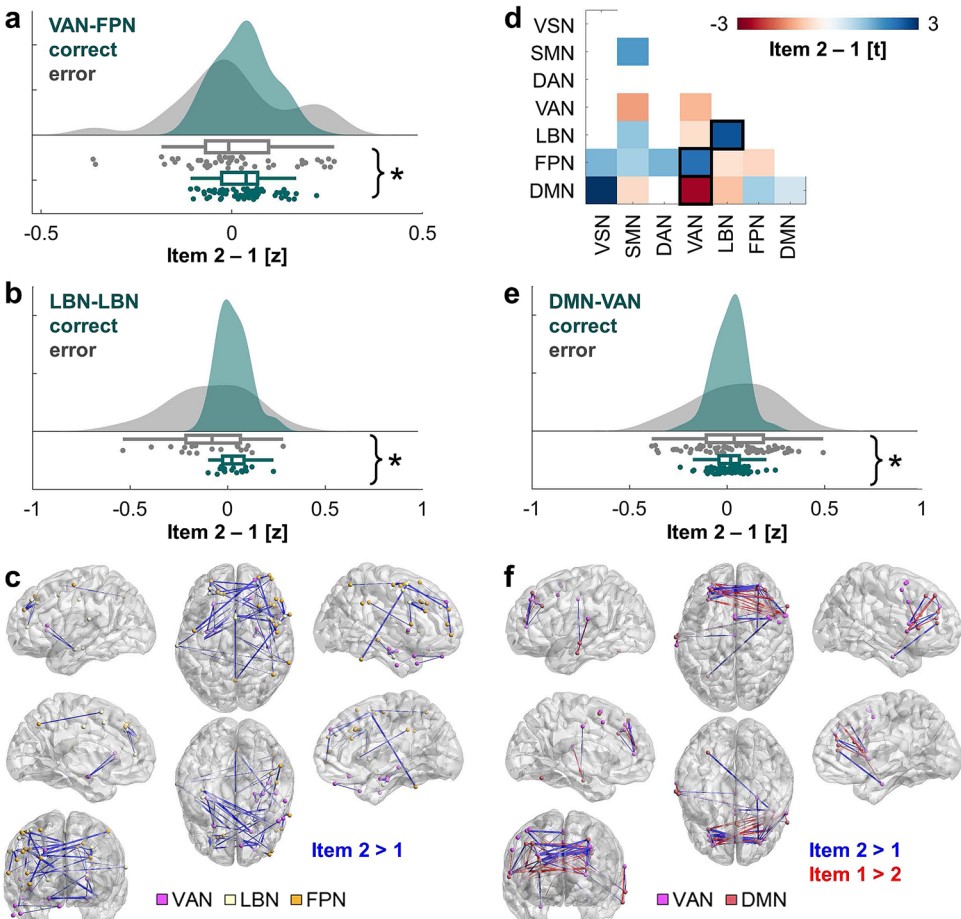

**Fig. 3 | Inter-regional theta information accumulation effects supporting successful WM. a, b** Successful CL performance was linked to increased theta synchrony to the second over first item between the VAN and FPN ($t(88) = 2.27$, $p = 0.026$) (**a**) and within the LBN ($t(36) = 2.62$, $p = 0.013$) (**b**). Data are represented as individual electrode-pair datapoints, and condition probability densities and medians calculated across electrode pairs from all subjects ($n = 67$ (**a**) or 19 (**b**) electrode pairs from 11 biologically independent samples). Boxplots present the medians and interquartile ranges, and whiskers the 1.5 × IQR from the quartile. Source data are provided as a Source Data file. Notes on *p*-values and network abbreviations are provided in the legend of **d**. *$p < 0.05$. **c** Electrode pairs contributing to the significant correct > error effects in **a** and **b**, overlaid on the MNI-152 template brain. The colors of electrodes indicate the network. Inter-electrode lines are shown in blue. Line thickness indicates the relative size of the item 2 > 1 difference ($0 < z < 0.25$), consistent with information accumulation. This figure was created using BrainNet Viewer[72]. **d** T-statistics from post hoc testing of accuracy across each pair of networks and within each network (FPN-VSN $t(8) = 1.39$, $p = 0.201$; DMN-VSN $t(2) = 2.96$, $p = 0.098$; SMN-SMN $t(55) = 1.72$, $p = 0.091$; VAN-

SMN $t(27) = -1.27$, $p = 0.217$; LBN-SMN $t(6) = 1.17$, $p = 0.286$; FPN-SMN $t(60) = 1.03$, $p = 0.307$; DMN-SMN $t(42) = -0.60$, $p = 0.554$; FPN-DAN $t(17) = 1.43$, $p = 0.172$; VAN-VAN $t(54) = -1.02$, $p = 0.315$; LBN-VAN $t(16) = -0.53$, $p = 0.604$; FPN-VAN $t(88) = 2.27$, $p = 0.026$; DMN-VAN $t(50) = -2.54$, $p = 0.014$; LBN-LBN $t(36) = 2.62$, $p = 0.013$; FPN-LBN $t(50) = -0.49$, $p = 0.627$; DMN-LBN $t(40) = -0.90$, $p = 0.376$; FPN-FPN $t(88) = -0.65$, $p = 0.517$; DMN-FPN $t(29) = 1.05$, $p = 0.301$; DMN-DMN $t(69) = 0.57$, $p = 0.571$). *P*-values are two-sided and uncorrected. VSN visual network, SMN somatomotor network, DAN dorsal attention network, VAN ventral attention network, LBN limbic network, FPN frontoparietal network, DMN default mode network; black box, $p < 0.05$. **e** Successful CL performance was linked to minimal change in theta synchrony between items in DMN-VAN interactions ($t(50) = -2.54$, $p = 0.014$; $n = 74$ electrode pairs from 11 biologically independent samples), same conventions as **a**, **b**. **f** Electrode pairs contributing to the significant correct < error effect in **e**, overlaid on the MNI-152 template brain. The colors of electrodes indicate the network. Inter-electrode lines are shown in blue (item 2 > 1) and red (item 1 > 2). Line thickness indicates the relative size of the difference ($-0.24 < z < 0.24$). This figure was created using BrainNet Viewer[72].

Collectively, these results are consistent with the hypothesis that sub-second changes in theta synchrony dually accumulate task-relevant information in and block irrelevant information from WM. Although theta synchrony is relatively stable during encoding on correct compared to error trials in DMN-VAN (Fig. 3e) and DMN-DAN (Fig. 4a) interactions, our results reveal that flexibly shifting DMN synchrony from the DAN to encode targets to the LBN to block distractors supports faster RT. Filtering is associated with rapid changes in theta synchrony between the DMN and other neocortical networks.

### Global theta network hub substrates of filtering

Having demonstrated changing patterns of theta synchrony to encode or filter in task-relevant information and block or filter out irrelevant information, we last sought to determine whether certain neocortical

networks were more globally synchronous during one state or the other. To determine the networks most correlated with RT on correct CF trials, we utilized graph theory. We used the node strength statistic (i.e., sum of unthresholded correlations across all electrode pairs including each electrode) to identify network hubs during filtering[58]. Summed correlation coefficients were then *z*-score normalized against null distributions, yielding one normalized node strength correlation per electrode. The linear mixed-effects model examining theta node strength by RT and brain network revealed an RT by network interaction ($F(6,206) = 2.88$, $p = 0.010$). The RT and network main effects were not significant ($p > 0.95$). Faster RT was linked to increased global theta synchrony in the DAN during the encoding of targets over distractors ($t(8) = -5.59$, $p = 0.0005$; Fig. 6a, b) and VAN during the blocking of distractors ($t(36) = 2.09$, $p = 0.043$; Fig. 6c, d). All post hoc

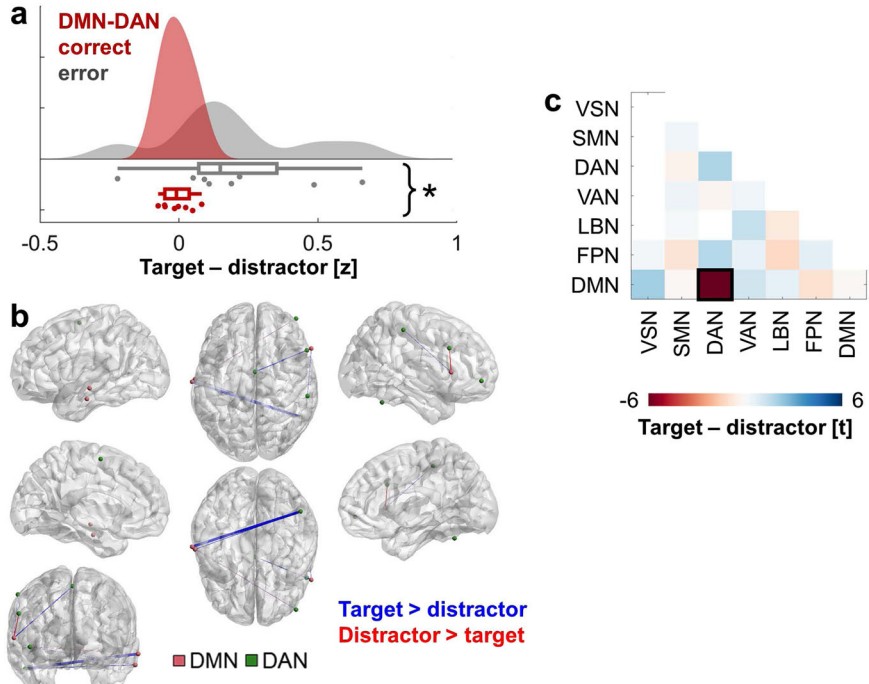

**Fig. 4 | Inter-regional theta filtering effects supporting successful WM.**
**a** Successful CF performance was linked to minimal change in theta synchrony between items in DMN-DAN interactions ($t(4) = -12.70$, $p = 0.0002$). Data are represented as individual electrode-pair datapoints, and condition probability densities and medians calculated across electrode pairs from all subjects ($n = 8$ electrode pairs from 11 biologically independent samples). Boxplots present the medians and interquartile ranges, and whiskers the $1.5 \times IQR$ from the quartile. Source data are provided as a Source Data file. Notes on $p$-values and network abbreviations are provided in the legend of **c**. *$p < 0.05$. **b** Electrode pairs contributing to the significant correct < error effect in **a**, overlaid on the MNI-152 template brain. The colors of electrodes indicate the network. Inter-electrode lines are shown in blue (target > distractor) and red (distractor > target). Line thickness indicates the relative size of the difference ($-0.32 < z < 0.32$). This figure was created using BrainNet Viewer[72]. **c** T-statistics from post hoc testing of accuracy across

each pair of networks and within each network (FPN-VSN $t(8) = 0.245$, $p = 0.813$; DMN-VSN $t(2) = 2.20$, $p = 0.158$; SMN-SMN $t(70) = 0.37$, $p = 0.714$; DAN-SMN $t(8) = -0.44$, $p = 0.675$; VAN-SMN $t(38) = 0.43$, $p = 0.673$; LBN-SMN $t(6) = 0.10$, $p = 0.920$; FPN-SMN $t(68) = -0.99$, $p = 0.326$; DMN-SMN $t(48) = -0.19$, $p = 0.848$; DAN-DAN $t(4) = -12.70$, $p = 0.0002$; VAN-DAN $t(4) = 1.95$, $p = 0.124$; FPN-DAN $t(24) = 1.84$, $p = 0.078$; DMN-DAN $t(4) = -12.70$, $p = 0.0002$; VAN-VAN $t(60) = 0.38$, $p = 0.702$; LBN-VAN $t(16) = 1.54$, $p = 0.144$; FPN-VAN $t(94) = 0.62$, $p = 0.536$; DMN-VAN $t(54) = 1.25$, $p = 0.216$; LBN-LBN $t(36) = -0.77$, $p = 0.449$; FPN-LBN $t(50) = -1.34$, $p = 0.188$; DMN-LBN $t(40) = 0.69$, $p = 0.497$; FPN-FPN $t(98) = 0.73$, $p = 0.469$; DMN-FPN $t(34) = -1.02$, $p = 0.314$; DMN-DMN $t(70) = -0.10$, $p = 0.920$). *P*-values are two-sided and uncorrected. VSN visual network, SMN somatomotor network, DAN dorsal attention network, VAN ventral attention network, LBN limbic network, FPN frontoparietal network, DMN default mode network; black box, $p < 0.05$.

---

results are provided in Fig. 6e. No effects in other networks reached significance ($p > 0.06$). We also calculated theta node strength statistics to examine behavioral accuracy by summing difference scores on correct and error CF and CL trials and generated normalized summed correlation coefficients on CL trials. No significant effects were detected in any model of these data ($p > 0.07$).

These results further support the hypothesis that sub-second changes in theta synchrony dually accumulate task-relevant information in and block irrelevant information from WM. Our results reveal that flexibly shifting global theta synchrony from the DAN to encode targets to the VAN to block distractors supports faster RT. This global shift implicates changing patterns of theta synchrony not only between the DMN and other networks (Fig. 5), but also in the overall engagement of the DAN during the gate open state and VAN during the gate closed state.

### Inter-regional alpha and beta mechanisms of information accumulation and filtering

There were more oscillatory electrodes in the theta band than either the alpha or beta band (Fig. S1e), which reflects the prevalence of theta oscillations during encoding and is a confound if comparing outcomes across frequencies. Thus, we submitted the alpha and beta synchrony data to the same single-subject and group-level analyses but caution against over-interpreting effects based on limited sampling.

In the alpha band ($n = 300$ electrode pairs; Fig. S3), the model examining RT and brain network on CL trials revealed a main effect of RT ($F(1,232) = 5.12$, $p = 0.025$), such that faster RT was linked to increased synchrony during the encoding of the second over first item. No significant effects were detected in any other model examining CL trials ($p > 0.07$). The model examining difference scores on CF trials as a function of accuracy and brain network revealed a network main effect ($F(33,232) = 2.51$, $p = 4 \times 10^{-5}$), but no accuracy main effect or interaction ($p > 0.07$). The equivalent model examining node strength revealed a network main effect ($F(6,86) = 5.31$, $p = 0.0001$) and an interaction ($F(6,86) = 2.76$, $p = 0.017$). The accuracy main effect was not significant ($p = 0.25$). Successful performance was linked to increased global synchrony during the encoding of targets over distractors in the FPN ($t(14) = 2.72$, $p = 0.017$). The model examining RT and brain network revealed an RT by network interaction ($F(33,232) = 1.58$, $p = 0.029$). Faster RT was linked to increased synchrony during the encoding of targets over distractors in LBN-SMN interactions ($t(2) = -10.02$, $p = 0.010$), and increased synchrony during the blocking of distractors in VAN-SMN ($t(12) = 5.34$, $p = 0.0002$), VAN-FPN ($t(6) = 8.28$, $p = 0.0002$), and LBN-DMN interactions ($t(16) = 2.66$, $p = 0.017$). The equivalent model examining node strength revealed an RT main effect ($F(6,86) = 6.22$, $p = 0.015$), such that faster RT was linked to increased global synchrony during the blocking of distractors. The network main effect and interaction were not significant ($p > 0.95$).

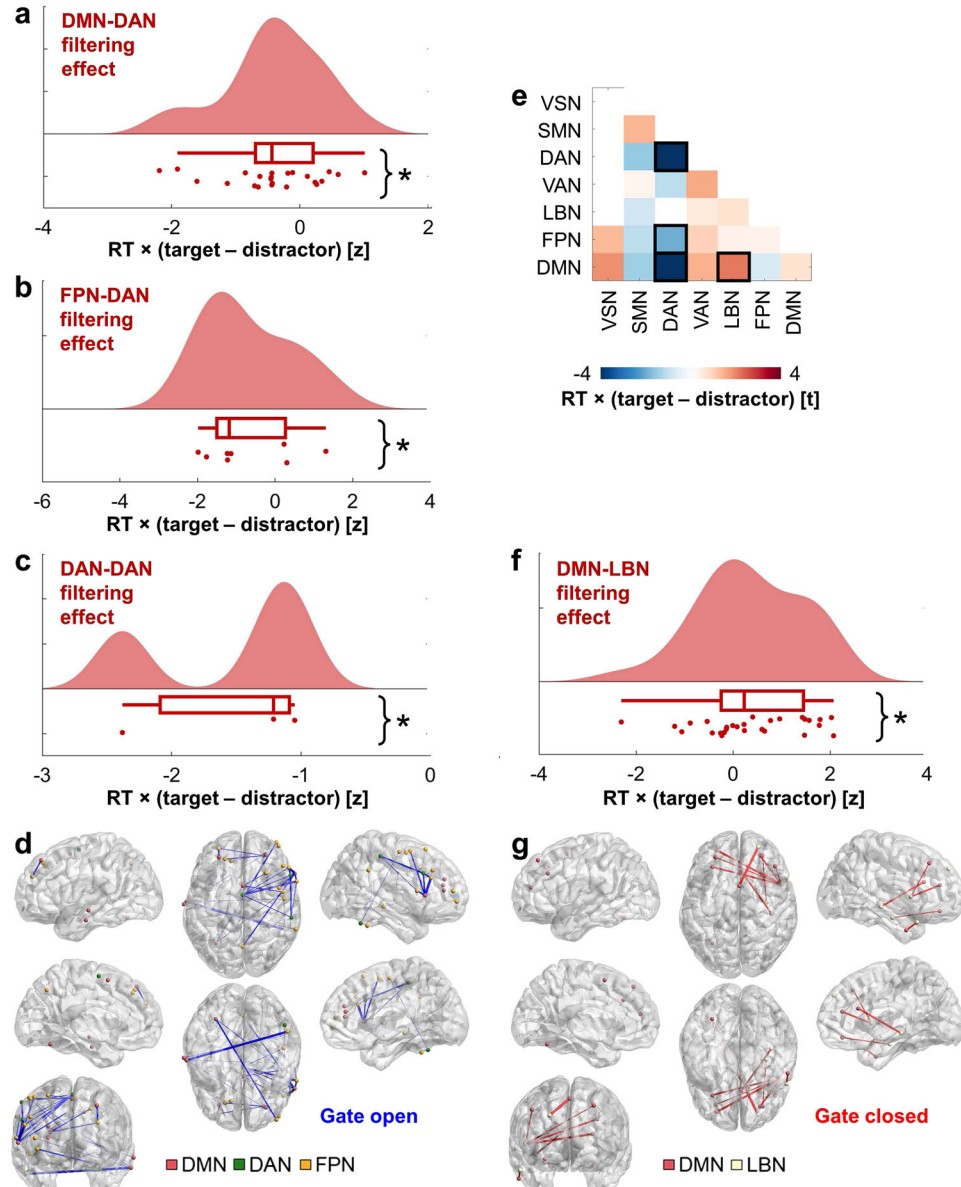

**Fig. 5 | Inter-regional theta filtering effects supporting efficient WM. a–c** On correct CF trials, faster behavioral RT was linked to increased theta synchrony to targets over distractors between the DMN and DAN ($t(4) = -8.55$, $p = 0.001$) (**a**) and FPN and DAN ($t(24) = -2.12$, $p = 0.045$) (**b**), and within the DAN ($t(4) = -5.14$, $p = 0.007$) (**c**). Data are represented as individual electrode-pair datapoints, and condition probability densities and medians calculated across electrode pairs from all subjects ($n = 8$ (**a**), 23 (**b**), or 3 (**c**) electrode pairs from 11 biologically independent samples). Boxplots present the medians and interquartile ranges, and whiskers the $1.5 \times$ IQR from the quartile. Source data are provided as a Source Data file. Notes on p-values and network abbreviations are provided in the legend of **e**. *$p < 0.05$. **d** Electrode pairs contributing to the significant RT correlation effect in **a–c** overlaid on the MNI-152 template brain. The colors of electrodes indicate the network. Inter-electrode lines are shown in blue. Line thickness indicates the relative size of the normalized RT × (target > distractor) correlation ($-2.20 < z < 0$), consistent with an open gate. This figure was created using BrainNet Viewer[72]. **e** T-statistics from post hoc testing of RT across each pair of networks and within each network (FPN-VSN $t(8) = 1.43$, $p = 0.190$; DMN-VSN $t(2) = 2.01$, $p = 0.182$; SMN-SMN $t(70) = 1.52$, $p = 0.134$; DAN-SMN $t(8) = -1.66$, $p = 0.135$; VAN-SMN $t(38) = 0.29$, $p = 0.771$; LBN-SMN $t(6) = -0.88$, $p = 0.413$; FPN-SMN $t(68) = -1.20$, $p = 0.234$;

DMN-SMN $t(48) = -1.57$, $p = 0.124$; DAN-DAN $t(4) = -5.14$, $p = 0.007$; VAN-DAN $t(14) = -1.19$, $p = 0.254$; FPN-DAN $t(24) = -2.12$, $p = 0.045$; DMN-DAN $t(4) = -8.55$, $p = 0.001$; VAN-VAN $t(60) = 1.67$, $p = 0.10$; LBN-VAN $t(16) = 0.55$, $p = 0.588$; FPN-VAN $t(94) = 1.07$, $p = 0.286$; DMN-VAN $t(54) = 1.55$, $p = 0.128$; LBN-LBN $t(36) = 0.72$, $p = 0.477$; FPN-LBN $t(50) = 0.397$, $p = 0.693$; DMN-LBN $t(40) = 2.28$, $p = 0.028$; FPN-FPN $t(98) = 0.33$, $p = 0.740$; DMN-FPN $t(34) = -0.75$, $p = 0.456$; DMN-DMN $t(70) = 0.73$, $p = 0.468$). $P$-values are two-sided and uncorrected. VSN visual network, SMN somatomotor network, DAN dorsal attention network, VAN ventral attention network, LBN limbic network, FPN frontoparietal network, DMN default mode network, black box, $p < 0.05$. **f** On correct CF trials, faster RT was linked to increased theta synchrony to distractors over targets between the DMN and LBN ($t(40) = 2.28$, $p = 0.028$; $n = 28$ electrode pairs from 11 biologically independent samples), same conventions as **a–c**. **g** Electrode pairs contributing to the significant RT correlation effect in **f** overlaid on the MNI-152 template brain. The colors of electrodes indicate the network. Inter-electrode lines are shown in red. Line thickness indicates the relative size of the normalized RT × (distractor > target) correlation ($0 < z < 2.24$), consistent with a closed gate. This figure was created using BrainNet Viewer[72].

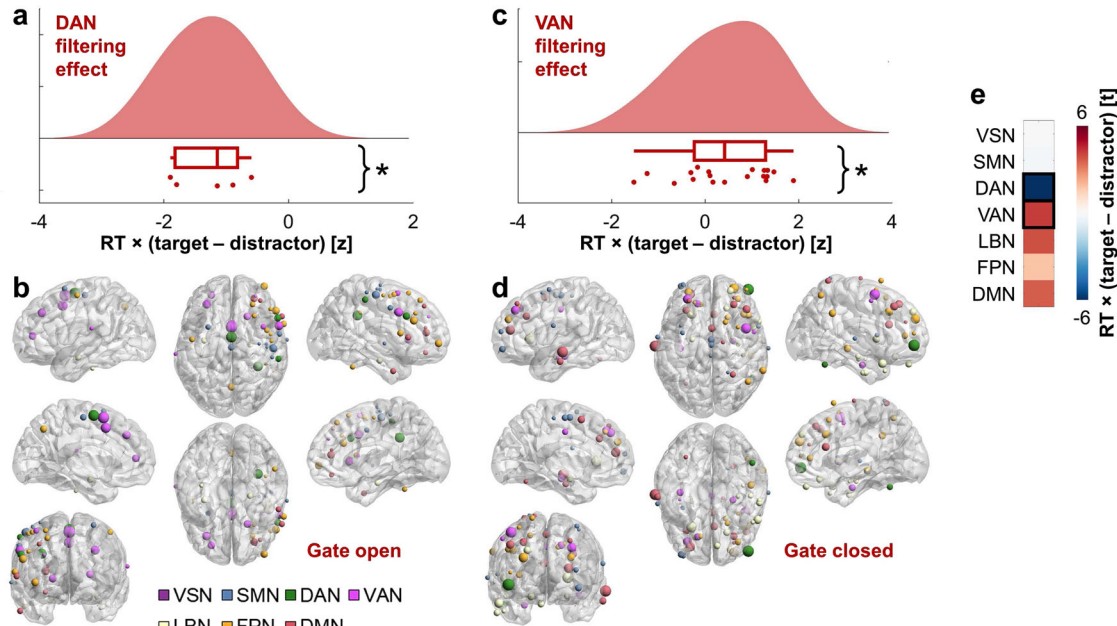

**Fig. 6 | Global theta network hub filtering effects supporting efficient WM. a** On correct CF trials, faster RT was linked to increased global theta synchrony with the DAN to targets over distractors ($t(8) = -5.59$, $p = 0.0005$). Data are represented as individual electrode datapoints, and condition probability densities and medians calculated across electrode pairs from all subjects ($n = 5$ electrodes from 11 biologically independent samples). The boxplot presents the median and interquartile range, and whiskers the $1.5 \times$ IQR from the quartile. Source data are provided as a Source Data file. Notes on p-values and network abbreviations are provided in the legend of **e**. *$p < 0.05$. **b** Normalized node strength correlations between theta synchrony difference scores on correct CF trials and RT where target > distractor synchrony preceded faster RT, across all subjects overlaid on the MNI-152 template brain. The colors of electrodes indicate the network. Electrode size indicates the relative size of the normalized summed RT × (target > distractor) correlation ($-2.22 < z < 0$), consistent with an open gate. This figure was created using BrainNet

Viewer[72]. **c** On correct CF trials, faster RT was linked to increased global theta synchrony with the VAN to distractors over targets ($t(36) = 2.09$, $p = 0.043$; $n = 19$ electrodes from 11 biologically independent samples), same conventions as **a**. **d** Same as **b** on correct CF trials with distractor > target synchrony. Electrode size indicates the relative size of the normalized summed RT × (distractor > target) correlation ($0 < z < 2.51$), consistent with a closed gate. **e** T-statistics from post hoc testing of RT per network (SMN $t(32) = -0.08$, $p = 0.940$; DAN $t(8) = -5.59$, $p = 0.0005$; VAN $t(36) = 2.09$, $p = 0.043$; LBN $t(32) = 1.94$, $p = 0.061$; FPN $t(56) = 0.89$, $p = 0.378$; DMN $t(42) = 1.83$, $p = 0.075$). P-values are two-sided and uncorrected. VSN visual network, SMN somatomotor network, DAN dorsal attention network, VAN ventral attention network, LBN limbic network, FPN frontoparietal network, DMN default mode network, black box, $p < 0.05$. This figure was created using BrainNet Viewer[72].

In the beta band ($n = 202$ electrode pairs; Fig. S4), the model examining difference scores on CL trials as a function of accuracy and brain network revealed a main effect of network ($F(31,138) = 1.54$, $p = 0.048$), but no accuracy main effect or interaction ($p > 0.24$). No significant effects were detected in other models examining CL trials ($p > 0.06$). The model examining difference scores on CF trials as a function of accuracy and brain network revealed a network main effect ($F(31,138) = 2.93$, $p = 9 \times 10^{-6}$) and an interaction ($F(31,138) = 1.65$, $p = 0.027$). The accuracy main effect was not significant ($p = 0.15$). Successful performance was linked to increased synchrony during the encoding of targets over distractors in LBN-SMN interactions ($t(2) = -5.57$, $p = 0.03$). No significant effects were detected in other models examining CF trials ($p > 0.10$).

## Discussion

We demonstrate cortical neurophysiological contributions to WM gating, not only maintenance, and identify both regional and inter-regional mechanisms characterizing cortical gating dynamics in the human brain. Our findings reveal that distributed HFB activities and theta oscillations govern neocortical network interactions which change dynamically as stimuli are presented in sequence, explaining accurate, fast, and flexible behavioral responses.

This study employed an established task of WM gating in which the position of the context in a sequence dictates whether subjects can adopt an input gating strategy to filter targets from distractors during encoding[1,13,14]. Superior behavioral performance on CF trials confirms that this manipulation was effective, consistent with previous reports.

Capitalizing on the unique single-trial and spatiotemporal precision of iEEG[23,24], we found that within-trial shifts in regional HFB activity and inter-regional theta synchrony supported input gating, both when selectively assigning information roles in memory during accumulation and when filtering targets over distractors. To do this, we tested within-trial shifts in iEEG measures as a function of subsequent behavioral accuracy and correlated these shifts with subsequent behavioral RT. We tested information accumulation by contrasting responses to the items presented before the context was known (CL), and filtering by contrasting responses to targets versus distractors when the context was known in advance (CF). We first demonstrated mechanisms of information accumulation on CL trials that are consistent with evidence from fMRI and EEG, and then applied the technique to identify mechanisms of filtering on CF trials.

On CL trials, because the context is not known in advance, items must be assigned a location that can be selectively accessed later (i.e., choosing the relevant target to influence responding and not the irrelevant one). This requires selective input gating that holds items in a role-addressable way[12,18,25]. Information accumulation-related increases in regional HFB activity were distributed across neocortical networks, and in inter-regional theta synchrony were distributed among FPN-VAN and intra-LBN interactions. CL effects predicted WM accuracy, suggesting that HFB and theta synchrony correlates of accumulating potentially relevant information in WM drive successful performance.

On CF trials, effects related to filtering in targets in HFB activity were distributed across neocortical networks, and in inter-regional

theta synchrony were distributed among DMN-DAN, FPN-DAN, and DAN-DAN interactions. In addition, theta synchrony shifted between encoding targets and blocking distractors, such that DMN-LBN synchrony was linked to filtering out distractors. Only the DMN was identified as a switch network, where changing patterns of theta synchrony with other neocortical networks supported both filtering in targets and filtering out distractors. Graph theoretical analysis further revealed that flexibly shifting global theta synchrony from the DAN to filter in targets to the VAN to filter out distractors supports faster behavioral responses. This global shift implicates changing patterns of theta synchrony not only between the DMN and other networks, but also in the overall magnitude of DAN engagement during the gate open state and VAN engagement during the gate closed state. CF effects predicted faster RT, suggesting that HFB and theta synchrony correlates of filtering drive efficient selection of relevant information from WM. Taken together, findings establish inter-regional theta synchrony as a core mechanism of flexible information encoding, and link information accumulation and filtering to dissociable brain networks by means of this core mechanism.

We provide evidence for a model in which neural oscillations dually accumulate relevant information in and block irrelevant information from WM[3,11,28]. Oscillations in the theta band served this dual role. Whereas converging evidence attributes gating to interactions between PFC and the striatum[1,2,12–15,18], our data implicate broader neocortical dynamics that also contribute to input gating and highlight a frequency-specific oscillatory mechanism that is distributed across brain networks. Indeed, theta synchrony has been proposed as a mechanism by which information is dynamically gated and transferred between PFC and distributed neocortical regions, enabling flexible behavior[3,29,30]. Accordingly, the present results extend a growing body of iEEG literature implicating inter-regional theta oscillations in attention and WM[23,24,31–37]. Our findings raise the intriguing possibility that theta oscillations support cognitive control by opening and closing gates of communication between regions, potentially analogous in function to the striatum but manifesting in DMN connectivity. Future empirical and theoretical work will be needed to better understand whether these broader network dynamics complement or are influenced by cortico-stratial interactions during gating.

The DMN is comprised of frontal, temporal, and parietal regions and characterized by relative quiescence in cognitive tasks[59]. Accordingly, we observed filtering-related changes in DMN synchrony without accompanying changes in regional HFB activity, which we contextualize based on the structural and functional and topology of the DMN. First, there is evidence from structural connectivity that DMN areas are densely connected, which could facilitate movement to many easily reachable states[60]. Second, the DMN connects brain areas associated with prediction, reward, episodic and semantic memory, action consideration, monitoring, and information integration, making it well-suited to act stochastically based on real, expected, and hypothetical outcomes[61]. Third, comparable to rest, the DMN is involved in tasks that demand more attention to internal thoughts than to the external environment, which may reflect its anatomical position farthest from sensorimotor cortices[62]. Fourth, task-negative hemodynamic responses in the DMN are not mirrored by decreased glucose metabolism, indicating energy consumption without fMRI activation[63]. Our results point to theta synchrony as a plausible mechanism of DMN function which moves the brain dynamically between goal-related states. This may be particularly the case when, in our task, the context must be internally maintained and used to guide the gate. Supporting this proposal, WM has been linked to intra-DMN[64] and DMN-FPN interactions[65]. Future work should examine how shifts among goal-related states, supported by the DMN, aids WM.

The DAN was globally synchronous in the theta band when filtering in targets and the VAN was globally synchronous in the theta band when filtering out distractors. The DAN is comprised of frontal and parietal regions and characterized by involvement in goal-directed attentional processes, whereas the VAN is comprised of frontal, temporal, and parietal regions and characterized by involvement in stimulus-driven attentional processes[66]. Our results point to theta synchrony as a plausible mechanism of DAN and VAN functions which connects brain areas as information is gated in WM. Assigning target to role-addressable locations in memory may be considered a goal-directed attentional process of the DAN and filtering out distractors may be considered a stimulus-driven attentional process of the VAN. Supporting this proposal, both DAN and VAN areas are situated between densely connected areas of the DMN and weakly connected areas of the FPN, facilitating integration and segregation across cognitive systems[60], and attentional reorienting has been linked to theta activity across both networks[67]. Future work should examine how attentional reorienting, supported by the DAN and VAN, aids WM.

Finally, we note caveats of this study. Electrode placement is determined solely by clinical needs, resulting in sparse sampling[23]. Intracranial placements tend to sample frontal, temporal, and parietal regions, as in this study. This limitation required us to collapse across broad networks, here, the Yeo-7 Atlas as opposed to the more spatially precise Yeo-17 Atlas[53], to achieve sufficient power for functional network analysis. Due to minimal sampling of occipital cortex, there were very few electrodes in the VSN, precluding observation of VSN effects. Furthermore, we observed that most task-active electrodes exhibited oscillations in the theta band, as opposed to alpha or beta band. Although this observation is consistent with research demonstrating theta oscillations across a more anterior anatomical distribution than alpha and beta oscillations[68], it nonetheless limited analysis of these frequency bands. On a related note, the Yeo-7 networks were defined based on group-level clustering analysis of resting-state fMRI data and not on the patients in this study. As a result, network boundaries may not correspond precisely with individual patients' network organization and did not account for overlapping networks[69]. Lastly, inter-regional synchrony may be mediated in part by subcortical structures, such as the striatum during gating. Although we capitalized on the unique capability of iEEG to investigate neocortical dynamics, it is nonetheless possible, and even likely, that theta oscillations further link cortical and subcortical networks on a whole-brain scale.

In summary, our findings address major questions in human neuroscience regarding how distributed brain networks transform incoming information into behavioral outputs[39]. We confirm that rapid neocortical mechanisms support the gating and accumulation of incoming information in WM. Our results establish inter-regional theta synchrony as a core mechanism of flexible information encoding and human behavior.

## Methods

### Subjects

Subjects were 11 right-handed adults ($M \pm SD$, $30 \pm 9$ years of age; 3 females) undergoing intracranial monitoring as part of clinical management of seizures. Demographic details are provided in Table S1. The initial sample included 13 patients selected based on above-chance task performance (inclusion criterion: $M < 0.35$ proportion errors; chance 0.5). Two of these patients were excluded based on other factors, one due to a lesion of the supplementary motor area and the other due to no iEEG sampling of frontal cortex. Neurosurgical patients were recruited from the University of California, Irvine (UCI), University of California, Davis (UCD), and California Pacific Medical Center (CPMC) hospitals. All patients provided written informed consent in accordance with the Declaration of Helsinki as part of the research protocol approved by the Institutional Review Board at each hospital.

### Working memory task

Subjects performed a single-trial WM task (Fig. 1a, b) adapted from fMRI research[1,13,14]. Three stimuli were presented sequentially on each

trial: one of two digits, one of two letters, and one of two symbols. The digit provided the context for the rule: digit 1 specified the symbol as the target and digit 2 specified the letter as the target. The two contexts, letters, and symbols were equiprobable and fully counterbalanced, as was the order of the letter and symbol items. The critical experimental manipulation was the position of the context. On context first (CF) trials, the context was the first of three stimuli. On context last (CL) trials, the context was the last of three stimuli. There were 40 CF and 40 CL trials presented in random order. Each stimulus was on screen for 500 ms, followed by a randomly jittered inter-stimulus fixation between 250 and 800 ms. WM was assessed in a two-alternative forced choice test. In this fourth and final stimulus event, response mappings were presented on the left and right sides of the screen. Only one mapping contained the target item for that trial, as determined by the context. Subjects indicated which side contained the target by pressing the F (left) or J (right) key. The response was self-paced. Trials were separated by a randomly jittered inter-trial fixation of at least 400 ms. Any trials with RT > 2.5 SD from a subject's mean were considered outliers and excluded from analysis. Data were collected using custom-built MATLAB (MathWorks Inc.) scripts with the Psychtoolbox-3 software extension.

## Electrode placement and localization

Macro-electrodes were surgically implanted for extra-operative recording based solely on the clinical needs of each patient. These electrodes were placed subdurally in 64-channel grids and 4- and 8-channel strips with 10-mm spacing (EcoG $n = 5$), or stereotactically in 8-, 10-, and 12-channel tracks with 5-mm spacing (sEEG $n = 6$). Anatomical locations were determined by co-registering post-implantation computed tomography (CT) coordinates to pre-operative magnetic resonance (MR) images, as implemented in FieldTrip[70]. Electrodes were localized in native space based on visual inspection of individual anatomy and transformed into standard MNI space for visualization across subjects. Electrodes were automatically localized into distinct networks based on their MNI coordinates according to the Yeo-7 Atlas[53] and verified by visual inspection. We selected this atlas because it has been widely used across neuroimaging modalities, including iEEG[71]. BrainNet Viewer was used to visualize electrodes and inter-electrode functional connections on the MNI-152 template brain[72].

## Data acquisition and preprocessing

Electrophysiological data were acquired using Nihon Kohden systems, sampled at 5 kHz (UCI), 1 kHz (CPMC), or 512 Hz (UCD). UCI data were resampled to 1 kHz. Raw data were filtered with 0.1-Hz high-pass and, if sampled at 1 kHz, 300-Hz low-pass finite impulse response filters, and 60-Hz line noise harmonics were removed using discrete Fourier transform. Continuous data were demeaned, epoched into trials (−1 s from the onset of the first stimulus to +1 s from the behavioral response), and manually inspected blind to electrode locations and experimental task parameters. Electrodes overlying seizure onset zones and electrodes and epochs displaying epileptiform activity or artifactual signal (from poor contact, machine noise, etc.) were excluded, ensuring that data considered for analysis represent healthy tissue[73]. Neighboring electrodes within the same anatomical structure were then bipolar montage referenced using consistent conventions (EcoG, anterior – posterior; sEEG, deep – surface)[31,32,51]. For EcoG grids, electrodes were referenced to neighboring electrodes on a row-by-row basis. An electrode was discarded if it did not have an adjacent neighbor, its neighbor was in a different anatomical structure, or both it and its neighbor were in white matter. Bipolar montages were used to minimize contamination from volume conduction, as is recommended for functional connectivity analysis[74]. Referenced data were manually re-inspected to reject any trials with residual noise. Error trials, except when comparing correct and error trials, and trials with outlier RTs were excluded from analysis[31,32,50,51]. The numbers of electrodes and trials included in iEEG analysis are provided in Table S1.

## High-frequency broadband activity

Power was computed from 70 to 150 Hz using a multitapering approach[75] in steps of 10 Hz (i.e., 70–80, 80–90... 140–150 Hz). Data segments were zero-padded to the next power of 2 to minimize filter-induced artifacts and the multitaper frequency spectrum was calculated by sliding a 300-ms window in 5-ms increments. An output resolution of 5 ms was used to equate the time axis across datasets sampled at 1 kHz and 512 Hz while preserving temporal precision. HFB activity was analyzed using statistical bootstrapping[35,48–52]. For each electrode and frequency, pre-stimulus (−250 to −50 ms from the onset of the first stimulus) raw power values were pooled into a single time series, from which 200 datapoints were randomly selected and averaged[51]. This step was repeated 1000 times to create bootstrapped distributions of pre-stimulus data. Post-stimulus raw power data were z-scored on the pre-stimulus distributions and then averaged across frequencies. This procedure adjusts the power outputs to correct for $1/f$ power scaling and reveals HFB activity that is elicited, with statistical significance, by the WM task.

Electrodes were considered responsive if HFB activity during target encoding or readout was above threshold ($z = 1.96$, equivalent to $\alpha = 0.05$, two tailed) for at least 10 consecutive timepoints (i.e., 50 ms). This duration criterion exceeds the minimum number of consecutive timepoints requisite for a time series to be considered significantly different from zero[76]. Target encoding was defined by time-locking CF and CL trials to the target stimulus and taking the trial-wise mean in a 750-ms epoch extending from the onset of the target through the minimum inter-stimulus fixation. Target readout was defined by time-locking trials to the behavioral response and taking the trial-wise mean in an epoch equal that subject's minimum RT, terminating at the onset of the response. Electrodes with mean HFB activity z-scores above 1.96 sustained for a minimum of 50 ms at any time in either epoch qualified as task-responsive and were included in further analysis[35,48,49]. This analysis is similar to methods used in single-unit neurophysiology and is naïve regarding electrode location and task condition.

## Oscillatory peak detection

Irregular-resampling auto-spectral analysis (IRASA) was used to disentangle true oscillatory components from the aperiodic $1/f$ slope[55]. Post-stimulus CF and CL data segments were epoched from the onset of the first stimulus to the onset of the test probe, zero-padded, and analyzed from 3 to 30 Hz. The IRASA method compresses and expands the epoched data with non-integer resampling factors to redistribute oscillatory components while leaving the aperiodic $1/f$ distribution intact. For each original and resampled data trace, the auto-spectrum was calculated using the fast Fourier transform after applying a Hanning window. The median was taken from the resampled auto-spectra to obtain the aperiodic $1/f$ component for each electrode and subtracted from the original power spectrum to isolate oscillatory residuals.

Peak detection was performed on the oscillatory residuals using statistical bootstrapping. For each electrode, the oscillatory residuals were pooled into a single frequency series, from which five datapoints were randomly selected and averaged. This step was repeated 1000 times to create bootstrapped distributions of frequency data. Oscillatory residuals were z-scored on the bootstrapped distributions and frequencies with prominence above threshold ($z = 1.96$, equivalent to $\alpha = 0.05$, two tailed) were considered peaks. Canonical frequency bands were used to classify peaks: theta (4–8 Hz), alpha (8–13 Hz), and beta (13-30 Hz). If more than one peak was detected within a frequency band, the largest was selected. This analysis isolates electrodes exhibiting frequency-specific oscillations prior to analysis of oscillatory phase-based functional connectivity.

## Functional connectivity

Oscillatory functional connectivity was quantified in single-trial epochs between all pairs of theta, alpha, and beta electrodes, respectively, per subject. Data segments were zero-padded and complex time series were calculated per-electrode at oscillatory peak frequencies (1/4 fractional bandwidth) using an adaptive, frequency-dependent sliding window of three cycles. Sliding windows were advanced in 5-ms increments and complex values were calculated using the fast Fourier transform after applying a Hanning window. Instantaneous phase values were extracted from each complex time series and phase-locking values (PLV) were calculated in electrode pairs[57]. The PLV method calculates the consistency in electrode-pair phase differences across a series of datapoints. Here, calculations were performed per-trial across 150 time points in two separate epochs extending 750 ms from stimulus onset (i.e., 750 ms at 5-ms resolution), and outputs were Fisher's Z-transformed. CF data were separately time-locked to the target and distractor stimuli and CL data to the first and second stimuli. We used the PLV method because it is widely used to quantify phase-based connectivity in electrophysiology data, including iEEG[31,32,58]. Critically, our analytic approach quantified changes in iEEG measures on a trial-by-trial basis (as described in "Subject-level brain-behavior relationships" section), and thus removed any idiosyncrasies of PLV such as zero-lag synchrony which do not vary systematically with task demands.

## Subject-level brain-behavior relationships

To address our primary question, we identified iEEG measures (i.e., HFB activity, PLV) in which within-trial shifts during encoding predicted subsequent behavioral accuracy and RT on a trial-by-trial basis. HFB data were averaged over the 750-ms epoch extending from the onset of each stimulus, yielding two datapoints per trial to match PLV outputs. On each trial, we calculated the difference in HFB activity or PLV between stimuli (CF, target − distractor; CL, second − first), and separately averaged the difference scores across correct and error trials per condition, and correlated difference scores with behavioral RT across correct trials per condition. Correlation analyses were performed using Spearman's rank partial correlation, $\rho_{XY \cdot Z}$, with $X$ set to the difference scores and $Y$ set to RT. To ensure that the difference between stimuli was not confounded by the length of the inter-stimulus fixation[77], $Z$ was set to the jitter and controlled for using partial correlation. We used correlation with RT to capture gating mechanisms without assuming that subjects employed gating strategies on all trials. Finally, for each electrode or electrode pair, the correlation was repeated 1000 times with trial numbers shuffled in the difference scores, generating a null distribution of correlations that could be expected by chance. Correlation coefficients were z-score normalized against null distributions to control for differences in trial counts between subjects (Table S1). These procedures yielded one mean correct difference score, one mean error difference score, and one normalized correlation coefficient per electrode or electrode pair, per condition.

The logic was three-fold. First, when target encoding is successful, the target is readily available at subsequent readout, operationalized by a faster correct response. On CF trials, faster RTs reflect proactive processes during encoding which flexibly 'gate in' the target and block the distractor. On CL trials, because RTs reflect reactive processes which 'gate out' whichever of two stimuli is deemed the target, faster RTs implicitly reflect processes related to the flexible encoding of two stimuli as distinct items[12,18,25]. Second, flexible encoding is operationalized as the difference in the strength of an iEEG measure between two stimuli within one trial (CF, target − distractor; CL, second − first). On CF trials, a positive difference and/or negative correlation with RT indicates that a greater target effect precedes a (faster) correct response, consistent with an 'open gate', and a negative difference

and/or positive correlation that a greater distractor effect precedes a (faster) correct response, consistent with a 'closed gate'. On CL trials, a positive difference and/or negative correlation indicates that a greater item 2 effect precedes a (faster) correct response, consistent with accumulating information in WM, and a negative difference and/or positive correlation that a greater item 1 effect precedes a (faster) correct response, consistent with a transition from encoding item 1 to item 2. Third, these procedures control for inter-individual variability in behavior (Fig. 1b, c) while mirroring the approach used in prior research[13]. By identifying mechanisms related to behavior in single trials, as opposed to mechanisms that differed between items across trials independent of behavior, these procedures isolate mechanisms of successful WM irrespective of omnibus behavioral differences between subjects. The use of Spearman's rank correlation accounted for an individual's tendency to respond more quickly or slowly.

Bayesian correlation further verified that inter-trial variability in the length of the inter-stimulus fixation did not drive inter-trial variability in behavioral RT[78]. The Bayes Factor (BF) quantifies both hypothesized and null effects, with $BF_{10} > 10$, $BF_{10} > 3$, and $BF_{10} > 1$ representing strong, modest, and anecdotal evidence of the hypothesized effect and $BF_{10} < 1/10$, $BF_{10} < 1/3$, and $BF_{10} < 1$ representing strong, modest, and anecdotal evidence of the null effect. Correlation was performed across CF and CL trials for each subject and all results supported the null effect ($BF_{10} < 0.51$). These null effects confirm that readout speed was not a byproduct of encoding time. Bayesian correlation was performed using the open-source JASP software (JASP Team, 2020).

## Node strength statistics

Graph theory was used to identify globally synchronous hubs during a given encoding state (CL, item 2 > 1 or item 1 > 2; CF, target > distractor or distractor > target). We used the node strength statistic as an unbiased measure reflecting the sum of all unthresholded connection weights (i.e., electrode-pair PLV difference scores and correlations) to each node (electrode). For each electrode, summed raw correlations were z-score normalized against the summed null distributions generated as described in "Subject level brain-behavior relationships" section[58]. These procedures yielded one summed correct difference score, one summed error difference score, and one normalized summed correlation per electrode, per condition.

## Group-level statistical models

All reported results are based on linear mixed-effects models of single-subject difference scores and normalized correlations, with subjects and nested electrodes as random intercepts[54]. For each iEEG measure and condition, we implemented one model examining difference scores with fixed effects of behavioral accuracy (i.e., correct vs. error) and brain network, and another model examining RT correlations with fixed effects of direction (i.e., normalized correlations vs. zero[35]) and brain network. Subjects and nested electrodes were modeled as random to indicate repeated measures and account for sources of inter-individual variability including electrode counts and sampling. Visual inspection of residual plots did not reveal any noticeable deviations from homoscedasticity or normality. The p-values of fixed effects and interactions were determined by F-tests calculated on the model outputs. In the case of significant interactions, post hoc models examined main effects per network, with either a fixed effect of behavioral accuracy on difference scores or direction on RT correlations. The p-values of fixed effects were determined by two-sided t-tests. Raincloud plots were used to visualize model results[79].

## Reporting summary

Further information on research design is available in the Nature Portfolio Reporting Summary linked to this article.

## Data availability

The data used in this study are available in the OSF database under accession code https://osf.io/rx2zd. Source data for Figs. 1b, c, 2a, c, 3a, b, e, 4a, 5a–c, f, and 6a, c are provided with this paper. Source data are provided with this paper.

## Code availability

The data and codes used in this study are available in the OSF database under accession code https://osf.io/rx2zd.

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

## Acknowledgements

We thank K. T. Jones, C. W. Hoy, B. J. Frick, N. Jaha, and R. T. Jimenez. This work was funded by grants from the National Institutes of Health (NINDS R00NS115918 to E.L.J.; NIMH R01MH111737 to M.D., R.T.K., and D.B.; NINDS R01NS021135 to R.T.K.; NINDS U19NS107609 to J.J.L. and R.T.K.) and Office of Naval Research (MURI N00014-16-1-2832 to D.B.).

## Author contributions

Conceptualization: E.L.J., M.D., R.T.K., and D.B.; Methodology: D.B.; Software: E.L.J.; Validation: E.L.J. and D.B.; Formal analysis: E.L.J.; Investigation: E.L.J. and I.S.; Resources: J.J.L., D.K.S., P.B.W., K.D.L., I.S., and F.G.; Data curation: E.L.J.; Writing – original draft: E.L.J. and D.B.; Writing – review and editing: E.L.J., M.D., R.T.K., and D.B.; Visualization:

E.L.J. and D.B.; Supervision: R.T.K. and D.B.; Project administration: M.D., R.T.K., and D.B.; Funding acquisition: E.L.J., M.D., R.T.K., and D.B.

## Competing interests

The authors declare no competing interests.
