## [Peer Review File · Nature Communications]

A rapid theta network mechanism for flexible information encodingREVIEWER COMMENTS

Reviewer #1 (Remarks to the Author):

Thank you for the opportunity to review “A rapid theta network mechanism for flexible information encoding” by Johnson and colleagues. The manuscript describes an iEEG investigation of how information is gated prior to encoding in working memory. The authors identify neocortical theta as the primary medium, particularly within the ventral attention and default mode networks.

The study addresses a clear and simple question. The evidence of a neocortical theta network that supports information gating could potentially be interesting to a wide audience and help to reconceptualize how information is routed in working memory. I commend the authors for separating the ephys activity into period and aperiodic components, and also for making the data openly available. I do, however, have some concerns and suggestions:

1. The main hypothesis is that there is – contrary to contemporary theories – an extrastriatal cortical network that contributes to gating. The authors certainly find evidence of some involvement of the theta network, but ultimately this is an incomplete answer to the original question because there is no data about striatal involvement. I don’t think this is a fatal flaw, but the narrative could perhaps be more nuanced to account for the fact that the contribution of striatal activity/synchrony is impossible to measure here.
2. Why did authors use PLV as opposed to another method that accounts for potentially superior zero-lag synchrony?
3. What do node sizes encode in the brain plots? E.g. Figure 3.
4. Why perform the RT analysis only on “task responsive” electrodes? This seems like a pre-selection step that only serves to double dip.
5. I did not understand why the authors chose to partition electrodes into oscillatory and non-oscillatory nodes. Surely all nodes have both oscillatory and non-oscillatory components, but in different proportions?

6. Much is made of correlations with behaviour, but we never see any data. The majority of figures are brain topographies.

7. "First, DMN areas are densely connected, which facilitates dynamic changes to many easily reachable states". I don't think this is true – the DMN cannot be identified using any data-driven clustering/community detection/ICA analysis in structural connectivity data.

Reviewer #2 (Remarks to the Author):

Johnson and colleagues used intracranial EEG recordings from neocortical sites in combination with an established working memory task to test a model in which neural oscillations dually accumulate information in and block irrelevant information from working memory. The authors report neocortical contributions to working memory that go beyond maintenance and include gating mechanisms that have been previously mainly assigned to the striatum. The authors provide evidence for a key role of cortical theta oscillations in information accumulation and gating mechanisms. Moreover, their findings provide insight into the involved network mechanisms. Specifically, they show that while the ventral attention network was implicated in both information accumulation and filtering mechanisms, filtering additionally and uniquely recruited the default mode network.

Overall, this is a well-designed study that leverages the unique opportunities that come with intracranial EEG in terms of spatial and temporal resolution. The findings are interesting and may contribute to our understanding of how working memory processes are implemented in the human brain. Although I value to potential contribution of this study, I have identified several issues that need to be carefully addressed.

(1) My main concern relates to the correlational analysis. The authors have performed a number of correlations, for quite a number of electrodes and each for CL and CF trials. The high number of correlations performed leads to an increased risk of false positives and I could not find how the authors corrected for the number of correlations performed. I may be wrong but it seemed to me as if the authors first tested how many electrodes showed a significant correlation with RTs in CF or CR trials and then focussed only on these electrodes, testing whether correlations for this subset of electrodes were positive or negative. If this was the case, I'm not convinced this is justified. In my view, the authors need to correct for the number for correlations performed and this correction should not be limited to those that (individually) reached significance. As the correlational analysis are at the very heart of the interpretations, I consider the appropriate correction for the number of tests performed critical.

(2) On a related note, the authors provide only relatively limited information on the magnitude, distribution and statistical significance of the correlations (mainly a range in the legends of Figs 3 and 4

and different line thickness reflecting correlation strength). The authors should add, for instance, (supplementary) tables that indicate the number of correlations performed, the actual size of the correlations and the significance level (after correction for multiple comparison, see my previous comment).

(3) While the authors focussed in their analyses mainly on RTs, it would be very interesting to see whether there are also links to the actual accuracy of the responses. Were iEEG parameters also linked to response accuracy? These additional analyses could be presented in the supplement.

MINOR COMMENTS

(4) While the number of errors was significantly lower for CF compared to CL trials, these trials types did not differ in RTs ($p=.107$). However, in the legend of Fig 1C it suggested that participants were faster on CF vs. CL trials. Similarly, at the beginning of the discussion (p. 13), it is suggested that there was “superior behavioral performance on CF trials”. These statements need to be corrected and it should be made explicit that while the response accuracy was higher in CF vs. CL trials, there was no reliable difference between trial types for RTs.

(5) In Figures 2a and b, the grey dots indicating electrodes that showed no response/no oscillations are difficult to read (against the grey brain image). I would recommend to use a different color for these dots.

Reviewer #1 (Remarks to the Author):

Thank you for the opportunity to review “A rapid theta network mechanism for flexible information encoding” by Johnson and colleagues. The manuscript describes an iEEG investigation of how information is gated prior to encoding in working memory. The authors identify neocortical theta as the primary medium, particularly within the ventral attention and default mode networks.

The study addresses a clear and simple question. The evidence of a neocortical theta network that supports information gating could potentially be interesting to a wide audience and help to reconceptualize how information is routed in working memory. I commend the authors for separating the ephys activity into period and aperiodic components, and also for making the data openly available. I do, however, have some concerns and suggestions:

Thank you for the positive evaluation of our study and providing important feedback to improve its suitability for publication. Point-by-point replies follow.

1. The main hypothesis is that there is – contrary to contemporary theories – an extrastriatal cortical network that contributes to gating. The authors certainly find evidence of some involvement of the theta network, but ultimately this is an incomplete answer to the original question because there is no data about striatal involvement. I don't think this is a fatal flaw, but the narrative could perhaps be more nuanced to account for the fact that the contribution of striatal activity/synchrony is impossible to measure here.

We absolutely agree; iEEG rarely, if ever, samples the striatum. This is stated in the revised Introduction (2nd paragraph).

In the Discussion, we describe the present results as evidence of neocortical involvement in gating beyond cortico-striatal interactions, not instead of cortico-striatal interactions. Future work is needed to understand whether neocortical mechanisms complement or are influenced by cortico-striatal interactions (4th paragraph). Indeed, we propose that theta oscillations link cortical and subcortical structures such as the striatum on a whole-brain scale (7th paragraph).

2. Why did authors use PLV as opposed to another method that accounts for potentially superior zero-lag synchrony?

We opted to use the PLV method because it is very widely used to quantify phase-based connectivity in electrophysiology data, including iEEG studies by our group and others (e.g., refs. 31, 32, 58), placing our results in the context of 20+ years of literature. Critically, our analytic approach quantified changes in iEEG measures on a trial-by-trial basis (e.g., target – distractor PLV difference scores, correlation between PLV difference scores and RT), and thus removed any idiosyncrasies of PLV such as zero-lag synchrony which do not vary systematically with task demands. This rationale is provided in the revised Methods (Functional connectivity).

3. What do node sizes encode in the brain plots? E.g. Figure 3.

Please note that we revised all figures to address point #6 about showing data, Reviewer 2's point #2 to provide information about individual correlations, and feedback from the editor to display individual datapoints. The revised figures show data as: 1) raincloud plots, which display individual datapoints, and condition probability densities and medians calculated across subjects (Fig. 1b-c), electrodes from all subjects (Fig. 2a, 2c, 6a, 6c), or electrode pairs from all subjects (Fig. 3a, 3b, 3e, 4a, 5a-c, 5f); and 2) brain topographies (Fig. 2b, 2d, 3c, 3f, 4b, 5d, 6b, 6d, S1a-d, S2-S4).

To answer the reviewer's question, nodes vary in size in the revised Fig. 2 plotting high-frequency broadband (HFB) activity and Fig. 6 plotting global theta network hub synchrony, a new graph theoretical analysis. In both figures, nodes represent individual electrodes, and the size of each electrode reflects the relative size of the subject-level statistic contributing to a given significant effect. In Fig. 2b, electrode size indicates the relative size of the item 2 > 1 difference in HFB activity on correct trials ($0 < z < 6.75$). In Fig. 2d, electrode size

indicates the relative size of the normalized RT × (target > distractor) correlation in HFB activity ($-3.16 < z < 0$). In Fig. 6b, electrode size indicates the relative size of the normalized summed RT × (target > distractor) correlation in theta synchrony ($-2.22 < z < 0$). In Fig. 6d, electrode size indicates the relative size of the normalized summed RT × (distractor > target) correlation in theta synchrony ($0 < z < 2.51$). This information is provided in the revised figure legends.

4. Why perform the RT analysis only on “task responsive” electrodes? This seems like a pre-selection step that only serves to double dip.

This is an important point. It is very common in iEEG research to first detect task-responsive electrodes based on HFB activity above baseline, blind to experimental parameters that might risk double dipping, and then include only task-responsive electrodes in further analysis (e.g., refs. 35, 48-52). This procedure is akin to spike detection in single-unit neuronal data and serves to remove quiet and ambiguous ‘noise’ from the data (e.g., Buzsáki et al., 2012; Lundqvist et al., 2016). This is stated in the revised Methods (High-frequency broadband activity, 2nd paragraph).

As noted above, this procedure should be performed blind to experimental parameters such as electrode location and task condition, and on data with unambiguous interpretation. For our study, that means we performed the procedure on HFB activity in all electrodes from both context-first (CF) and context-last (CL) trials during the encoding and retrieval of target items that were correctly identified at test. By analyzing CF and CL trials, the analysis was blind to the experimental parameter of condition. By analyzing the target and retrieval epochs of correct trials, we sought to select electrodes responsive to stimuli that were seen, encoded, and retrieved. At the most basic level, the subject’s task was to encode and retrieve the target stimulus and, thus, this procedure operationalized task-responsiveness. As reported in the Results (High-frequency broadband mechanism of information encoding), our procedure yielded a percentage of task-responsive electrodes that is consistent with literature on task responsiveness across a range of cognitive tasks and brain regions.

In contrast, correlations with RT were performed per condition, using difference scores (CF, target – distractor regardless of presentation order; CL, item 2 – 1 regardless of target position). Thus, we adopted established methods and did not double dip.

Additional references:

Buzsáki, G., Anastassiou, C.A., & Koch, C. The origin of extracellular fields and currents—EEG, ECoG, LFP and spikes. *Nat Rev Neurosci* 13, 407–420 (2012).

Lundqvist, M., Rose, J., Herman, P., Brincat, S. L., Buschman, T. J., & Miller, E. K. Gamma and beta bursts underlie working memory. *Neuron* 90, 1-13 (2016).

5. I did not understand why the authors chose to partition electrodes into oscillatory and non-oscillatory nodes. Surely all nodes have both oscillatory and non-oscillatory components, but in different proportions?

We agree that all electrophysiological data have non-oscillatory components, such as the ubiquitous aperiodic 1/f-like slope. However, not all electrophysiological data have robust oscillatory components, or robust oscillatory components at multiple frequencies. For this reason, it is important to demonstrate that a frequency-specific oscillation is present prior to analysis of frequency-specific oscillatory phase, as in analysis of phase-based connectivity. We adopted best practices to isolate electrodes exhibiting a robust oscillatory component in each frequency band (theta, alpha, beta) prior to analyzing phase-based connectivity in that frequency band. For analysis of theta phase-based connectivity, we first isolated electrodes displaying an oscillation in the theta band and then extracted phase at peak oscillatory theta frequency. We repeated these steps for analysis of alpha and beta phases, respectively. These steps ensured that all analyses of phase-based connectivity reflect underlying physiology (i.e., oscillations). This is stated in the revised Methods (Oscillatory peak detection, 2nd paragraph).

Note that PLV is computed from the phase of the data and does not concern non-oscillatory components like signal amplitude. Had we computed PLV at non-oscillatory electrodes and frequencies, the outputs would have been ambiguous to interpret. In contrast, for analysis of HFB activity, which is computed from the amplitude of the data and does not concern phase, we analyzed all electrodes regardless of oscillatory components.

References (list is not exhaustive):

Caplan, J. B., Bottomley, M., Kang, P. & Dixon, R. A. Distinguishing rhythmic from non-rhythmic brain activity during rest in healthy neurocognitive aging. *NeuroImage* 112, 341–352 (2015).

Donoghue, T. et al. Parameterizing neural power spectra into periodic and aperiodic components. *Nat Neurosci* 23, 1655–1665 (2020).

Hughes, A. M., Whitten, T. A., Caplan, J. B. & Dickson, C. T. BOSC: a better oscillation detection method, extracts both sustained and transient rhythms from rat hippocampal recordings. *Hippocampus* 22, 1417–1428 (2012).

Watrous, A. J., Miller, J., Qasim, S. E., Fried, I. & Jacobs, J. Phase-tuned neuronal firing encodes human contextual representations for navigational goals. *eLife* 7, e32554 (2018).

Wen, H. & Liu, Z. Separating fractal and oscillatory components in the power spectrum of neurophysiological signal. *Brain Topogr* 29, 13–26 (2016).

6. Much is made of correlations with behaviour, but we never see any data. The majority of figures are brain topographies.

Agreed. Among the figure revisions noted above, we include figures showing all brain-behavior relationships. Specifically, iEEG measures were correlated with trial-by-trial behavior per subject, as described in the revised Methods (Subject-level brain-behavior relationships). Note that in the revision, we z-score normalized the correlations against null distributions to control for differences in trial counts between subjects for group-level analyses and display in figures. Normalized correlation data are shown in the revised Fig. 2c-d on HFB activity, Fig. 5 on theta synchrony, and Fig. 6 on global theta network hub synchrony. Each figure follows the same design of raincloud plot of all individual correlations contributing to a significant effect and accompanying plot of the same data on the brain. In brain topographies, the size of each electrode (Fig. 2d, 6b, 6d) or electrode pair (Fig. 5d, 5g) reflects the relative size of the subject-level correlation contributing to a given significant effect. All correlations are provided in Fig. S2e-f, S3e-f, and S4e-f.

7. “First, DMN areas are densely connected, which facilitates dynamic changes to many easily reachable states”. I don’t think this is true – the DMN cannot be identified using any data-driven clustering/community detection/ICA analysis in structural connectivity data.

The quoted sentence references Gu et al., 2015. Their abstract states, “Our results suggest that densely connected areas, particularly in the default mode system, facilitate the movement of the brain to many easily reachable states.” The authors made this claim based on analysis of structural connectivity data (i.e., diffusion tractography), and we don’t have reason to question their work. Nonetheless, we rephrased the sentence to make clear that there is evidence for this, rather than suggesting it is fully established. If this (or a similar organization) is the case, we propose that theta synchrony provides a plausible *functional* mechanism enabling the DMN to move between easily reachable states, as constrained by structural connectivity. In the revision, we rephrased the sentence to, “First, there is evidence from structural connectivity that DMN areas are densely connected, which could facilitate movement to many easily reachable states” (Discussion 5th paragraph).

We will correct our claim if we have misinterpreted the study by Gu et al.

Reference:

Gu, S. et al. Controllability of structural brain networks. *Nat Commun* 6, 8414 (2015).

Reviewer #2 (Remarks to the Author):

Johnson and colleagues used intracranial EEG recordings from neocortical sites in combination with an established working memory task to test a model in which neural oscillations dually accumulate information in and block irrelevant information from working memory. The authors report neocortical contributions to working memory that go beyond maintenance and include gating mechanisms that have been previously mainly assigned to the striatum. The authors provide evidence for a key role of cortical theta oscillations in information accumulation and gating mechanisms. Moreover, their findings provide insight into the involved network mechanisms. Specifically, they show that while the ventral attention network was implicated in both information accumulation and filtering mechanisms, filtering additionally and uniquely recruited the default mode network. Overall, this is a well-designed study that leverages the unique opportunities that come with intra-cranial EEG in terms of spatial and temporal resolution. The findings are interesting and may contribute to our understanding of how working memory processes are implemented in the human brain. Although I value to potential contribution of this study, I have identified several issues that need to be carefully addressed.

Thank you for the positive evaluation of our study and providing important feedback to improve its suitability for publication. Point-by-point replies follow.

(1) My main concern relates to the correlational analysis. The authors have performed a number of correlations, for quite a number of electrodes and each for CL and CF trials. The high number of correlations performed leads to an increased risk of false positives and I could not find how the authors corrected for the number of correlations performed. I may be wrong but it seemed to me as if the authors first tested how many electrodes showed a significant correlation with RTs in CF or CR trials and then focussed only on these electrodes, testing whether correlations for this subset of electrodes were positive or negative. If this was the case, I'm not convinced this is justified. In my view, the authors need to correct for the number for correlations performed and this correction should not be limited to those that (individually) reached significance. As the correlational analysis are at the very heart of the interpretations, I consider the appropriate correction for the number of tests performed critical.

We appreciate the reviewer's concern about multiple comparisons and thought carefully about how to address it. Indeed, despite 40-50% of individual electrodes being identified as part of electrode pairs with significant theta synchrony x RT correlations, the percentage of individual electrode pairs was relatively low. Correcting for multiple comparisons would have further lowered this percentage. Instead, we opted to perform an additional set of group-level statistical analyses on all correlation data that did not require multiple tests. This resulted in a substantially revised manuscript, where results reflect all data and, critically, reveal both default mode and attention network mechanisms of filtering, strengthening our original submission. We adopted complementary methods from two influential iEEG studies, Kam et al., 2019, and Solomon et al., 2017, which we describe below.

In Kam et al., instead of identifying functional connectivity-behavior correlations in individual electrode pairs and then localizing electrodes contributing to electrode pairs with significant correlations to brain networks, the authors grouped electrodes to networks and then tested for significant correlations across the entire network. By testing the entire network once instead of testing however many electrode pairs however many times, this approach obviates the need to correct for multiple comparisons. They used a group-level, linear mixed-effect model comparing correlations to zero, with subjects modeled as random effects to account for differences in electrode sampling between subjects. We adopted this approach. By comparing all correlations, both negative and positive, to zero, our new analyses reveal whether there is directionality (e.g., target > distractor or distractor > target synchrony preceding faster behavioral RT). We used linear mixed-effects models with direction (i.e., correlations vs. zero) and brain network as fixed effects. In the case of significant interactions, post hoc models tested for a direction effect per network. This is detailed in the revised Methods (Group-level statistical models).

We also drew from Solomon et al., who computed theta synchrony between all pairs of electrodes per subject as a function of behavioral accuracy (i.e., subsequently recalled vs. not recalled words in a long-term memory task) and revealed increased global synchrony during the encoding of subsequently recalled words. We sought to adopt their procedures for normalizing subject-level statistics prior to group-level analysis and plotting.

Because subjects tend to recall more words than not, the authors z-score normalized the outputs against null distributions generated by shuffling recalled/not recalled labels prior to group-level analysis. We adopted a similar approach by z-score normalizing all correlations against null distributions to control for differences in trial counts between subjects prior to group-level analysis. This procedure also normalized the data for plotting on the group level in all revised figures. This is detailed in the revised Methods (Subject-level brain-behavior relationships).

These new analyses addressed the multiple-comparison problem by determining statistical significance from group-level analysis of all data at once. Importantly, the new results did not qualitatively change our conclusions about theta synchrony mechanisms supporting filtering on CF trials. Rather, we believe our conclusions are strengthened. For instance:

In the analysis of high-frequency broadband (HFB) activity, our model revealed a significant direction main effect, such that faster RT was linked to increased HFB activity during the encoding of targets over distractors. Full results are provided in the revised Results (High-frequency broadband mechanism of information encoding) and Fig. 2c-d.

In the analysis of theta synchrony, our model revealed a significant direction by brain network interaction, such that faster RT was linked to increased target > distractor synchrony in some networks (e.g., DMN-DAN) and increased distractor > target synchrony in other networks (DMN-LBN). Full results are provided in the revised Results (Theta network mechanism of filtering) and Fig. 5.

References:

Kam, J. W. Y. et al. Default network and frontoparietal control network theta connectivity supports internal attention. *Nat Hum Behav* 3, 1263-1270 (2019).

Solomon, E. A. et al. Widespread theta synchrony and high-frequency desynchronization underlies enhanced cognition. *Nat Commun* 8, 1704 (2017).

(2) On a related note, the authors provide only relatively limited information on the magnitude, distribution and statistical significance of the correlations (mainly a range in the legends of Figs 3 and 4 and different line thickness reflecting correlation strength). The authors should add, for instance, (supplementary) tables that indicate the number of correlations performed, the actual size of the correlations and the significance level (after correction for multiple comparison, see my previous comment).

We revised all figures to address this point to provide information about individual correlations, Reviewer 1's point #6 about showing data, and feedback from the editor to display individual datapoints.

Revised figures show data as: 1) raincloud plots, which display individual datapoints, and condition probability densities and medians calculated across subjects (Fig. 1b-c), electrodes from all subjects (Fig. 2a, 2c, 6a, 6c), or electrode pairs from all subjects (Fig. 3a, 3b, 3e, 4a, 5a-c, 5f); and 2) brain topographies (Fig. 2b, 2d, 3c, 3f, 4b, 5d, 5g, 6b, 6d, S1a-d, S2-S4).

Each figure follows the same design of raincloud plot of all individual correlations contributing to a significant effect and accompanying plot of the same data on the brain. In brain topographies, the size of each electrode (Fig. 2d, 6b, 6d) or electrode pair (Fig. 5d, 5g) reflects the relative size of the subject-level correlation contributing to a given significant effect. All correlations are provided in Fig. S2e-f, S3e-f, and S4e-f.

The revised Results indicate the number of electrodes or electrode pairs included in each analysis. HFB activity n = 145 electrodes, theta synchrony n = 110 electrodes and 1,374 electrode pairs, alpha synchrony n = 50 electrodes and 300 electrode pairs, and beta synchrony n = 34 electrodes and 202 electrode pairs.

(3) While the authors focussed in their analyses mainly on RTs, it would be very interesting to see whether there are also links to the actual accuracy of the responses. Were iEEG parameters also linked to response

accuracy? These additional analyses could be presented in the supplement.

We agree completely. As part of our substantially revised submission, we include an entirely new set of subject- and group-level analyses of behavioral accuracy which mirror the approach of correlating iEEG measures with RT. On the subject level, instead of correlating difference scores (CF, target – distractor; CL, item 2 – 1) with RT on correct trials, we averaged the difference scores separately for correct and error trials. On the group level, instead of comparing correlations to zero, we compared correct and error trials. These analyses are detailed in the revised Methods (Subject-level brain-behavior relationships, Group-level statistical models).

These new analyses revealed HFB activity and theta synchrony mechanisms supporting information accumulation on CL trials.

In the analysis of HFB activity, our model revealed a significant accuracy main effect, such that successful performance was linked to increased HFB activity during the encoding of the second over first item. Full results are provided in the revised Results (High-frequency broadband mechanism of information encoding) and Fig. 2a-b.

In the analysis of theta synchrony, our model revealed a significant accuracy by brain network interaction, such that successful performance was linked to increased item 2 > 1 synchrony in certain networks (e.g., VAN-FPN) but minimal change between items in other networks. Full results are provided in the revised Results (Theta network mechanism of information accumulation, Theta network mechanism of filtering) and Fig. 3 and 4.

To provide information on the magnitude, distribution, and statistical significance of the accuracy effects (point #2), the corresponding revised figures follow the same design of raincloud plot of all individual correct and error difference scores contributing to a significant effect and accompanying plot of the same data on the brain. In brain topographies, the size of each electrode (Fig. 2b) or electrode pair (Fig. 3c, 3f, 4b) reflects the relative size of the subject-level difference contributing to a given significant effect. All difference scores are provided in Fig. S2a-d, S3a-d, and S4a-d.

MINOR COMMENTS

(4) While the number of errors was significantly lower for CF compared to CL trials, these trials types did not differ in RTs ($p=.107$). However, in the legend of Fig 1C it suggested that participants were faster on CF vs. CL trials. Similarly, at the beginning of the discussion (p. 13), it is suggested that there was “superior behavioral performance on CF trials”. These statements need to be corrected and it should be made explicit that while the response accuracy was higher in CF cs. CL trials, there was no reliable difference between trial types for RTs.

The plots showing performance accuracy and RT are separated in the revised Fig. 1. The revised legend of Fig. 1c states, “RT did not differ significantly between CF and CL trials.”

(5) In Figures 2a and b, the grey dots indicating electrodes that showed no response/no oscillations are difficult to read (against the grey brain image). I would recommend to use a different color for these dots.

These figures are re-plotted with the no-response and no-oscillation electrodes in a darker color to make them easier to read. Please note that they now appear in the revised Fig. S1.

REVIEWERS' COMMENTS

Reviewer #1 (Remarks to the Author):

The authors have comprehensively addressed my concerns and I recommend publication.

Reviewer #2 (Remarks to the Author):

The authors addressed my previous comments appropriately. I have non further comments and think that this manuscript will make a relevant contribution to the litreature on working memory mechanisms.

Reviewer 1:

The authors have comprehensively addressed my concerns and I recommend publication.

Reviewer 2:

The authors addressed my previous comments appropriately. I have no further comments and think that this manuscript will make a relevant contribution to the literature on working memory mechanisms.

We thank both reviewers for their feedback on our initial submission, and for endorsing our manuscript for publication in *Nature Communications*!